# Adverse Effects Due to the Use of Upper Limbs Exoskeletons in the Work Environment: A Scoping Review

**DOI:** 10.3390/biomimetics10050340

**Published:** 2025-05-21

**Authors:** Omar Flor-Unda, Rafael Arcos-Reina, Susana Nunez-Nagy, Bernardo Alarcos

**Affiliations:** 1Ingeniería Industrial, Facultad de Ingeniería y Ciencias Aplicadas, Universidad de Las Américas, Quito 170125, Ecuador; 2Escuela de Fisioterapia, Facultad de Ciencias de la Salud, Universidad de Las Américas, Quito 170125, Ecuador; rafael.arcos@udla.edu.ec; 3Department of Nursing and Physiotherapy, Faculty of Medicine and Health Sciences, University of Alcalá, 28805 Alcalá de Henares, Spain; susana.nunez@uah.es; 4Humanization in the Intervention of Physiotherapy for the Integral Attention to the People Group (HIPATIA) Group, University of Alcalá, 28805 Alcalá de Henares, Spain; 5Polytechnic School, University of Alcalá, 28805 Alcalá de Henares, Spain; bernardo.alarcos@uah.es

**Keywords:** upper-limb exoskeleton, exoskeleton device, musculoskeletal diseases, muscle fatigue, postural balance

## Abstract

Both for design issues and for the study, analysis, and understanding of the interaction of workers with exoskeletons, the study of adverse effects provides criteria to improve the design of more efficient exoskeletons with better ergonomics and long-term usability. In this work, a scoping review was carried out on adverse effects due to the prolonged use of upper-limb exoskeletons, which have been evidenced in the scientific literature. The causes of the effects are described in terms of their impacts on the physiological, psychological, and technological aspects that affect the user. A scoping review of articles of the last ten years on negative effects of upper-extremity exoskeletons for industrial tasks was carried out following the guidelines of the PRISMA^®^ methodology with three phases: formulation of questions, definition of scopes and exhaustive search in SCOPUS, Web of Science, Science Direct, Taylor & Francis, and PubMed. The selection was made by two review authors with a Cohen’s Kappa coefficient of 0.9530, indicating high agreement. The effectiveness of upper-limb exoskeletons depends on the environment and the task, so an adaptable ergonomic design, field validations, and standards are required to ensure their functionality and acceptance. Use of exoskeletons mainly activates the posterior deltoid and latissimus dorsi and reduces the activity of muscles such as the trapezius, pectoralis major, anterior and middle deltoids, biceps brachii, brachioradialis, and flexor carpi radialis.

## 1. Introduction

Despite the multiple developments in upper-limb exoskeletons (ULEs), there is a lack of systematized studies on the adverse effects of using exoskeletons. Studying the adverse effects in the work environment is important because it allows us to identify and understand the potential risks associated with their implementation beyond the widely studied benefits [1]. This critical perspective is essential to ensure these technologies’ safe, effective, and sustainable adoption in real-world tasks, where factors such as anthropometric fit, ergonomics, comfort, and user-device interaction can directly affect worker health, performance, and acceptance. Analyzing these effects contributes to designing regulations, protocols for use, and technological improvements, ensuring that exoskeletons reduce the physical load and minimize the appearance of new discomfort or injury, especially in prolonged or repetitive use [2].

One of the most important challenges in the development and evolution of exoskeletons is identifying and controlling the negative effects that prolonged use can cause during highly physically demanding tasks [3]. Using exoskeletons involves several direct effects on the wearer’s muscles, joints, and bones, which can range from improved biomechanics to potential long-term physiological adaptations [4]. The study of these effects is essential to optimize the design of exoskeletons, ensure user safety, and maximize functional benefits [5].

Positive and negative impacts of using exoskeletons in moderate-load handling tasks have been identified. The vast majority of studies that address exoskeletons have focused on the advantages that these devices provide in terms of reducing perceived physical exertion [6], reducing muscle load [7,8], improving posture and comfort, and increasing mobility [9]. In addition to helping individuals with spinal cord injury (SCI) regain the ability to ambulate, the rapidly evolving capabilities of robotic exoskeletons provide an array of secondary biophysical benefits that can reduce the complications resulting from pro-longed immobilization.

The benefits of increased lifelong overground walking capacity include improved upper body muscular fitness, circulatory response, regular bowel movement, and reduced pain and spasticity. Beyond the positive changes related to physical and biological function, exoskeletons have been suggested to improve SCI individuals’ quality of life (QOL) by allowing increased participation in day-to-day activities [2]. Most of the currently available studies that have reported on the impact of exoskeletons on the QOL and prevention of secondary health complications in individuals with SCI are small-scale and heterogeneous in nature. Moreover, few meta-analyses and reviews have attempted to consolidate the dispersed data to reach more definitive conclusions about the effects of exoskeleton use.

Adverse effects have been less studied in the scientific literature, with few studies addressing negative impacts on five fundamental aspects: physical discomfort and ergonomic limitations [10,11], increased cognitive load [8,12], redistribution of musculoskeletal load, usability and acceptance issues [11,13,14], and increased physical strain in unsupported areas. Figure 1 illustrates a bibliometric review based on the occurrence of terms related to studies related to the effects of upper-limb exoskeletons.

Figure 1 highlights three main clusters. The first, in red, focuses on experimental studies with humans and terms such as “human experiment”, “task performance”, and “upper extremity”, which address how the use of exoskeletons can impact biomechanics and human performance, with possible adverse effects such as muscle overload. The green cluster examines specific muscle aspects, such as “skeletal muscle”, “electromyography (EMG)”, and “muscle activities”, reflecting research focused on muscle function and activity in the face of prolonged use of these devices. The blue cluster includes technical terms such as “exoskeleton (robotics)”, “upper-limb”, and “joints (anatomy)”, exploring the interaction between exoskeletons and the biomechanics of the upper-limb. According to Figure 1, the negative effects are related to factors such as exoskeleton design, excessive muscle load, and use in specific tasks, particularly in work environments.

Figure 2 presents a graph of the relative frequency with which studies that address the advantages and disadvantages in general of the use of upper limb exoskeletons are evidenced. Studies of the SCOPUS database have been obtained with the following text string: (“upper limb” OR “arm” OR “forearm” OR “hand”) AND (“exoskeleton” OR “wearable” OR “robotic” OR “assistive”) AND (“industrial” OR “workplace” OR “manufacturing” OR “factory”) AND (“workers” OR “ employees” OR “operators” OR “laborers”) AND (“ergonomics” OR “safety” OR “performance” OR “fatigue”) AND (“rehabilitation” OR “support” OR “augmentation” OR “enhancement”).

Figure 2 shows that in the studies of exoskeletons for upper limbs for industrial work, the advantages of the use of these devices are mentioned, while the disadvantages that include adverse or counterproductive effects due to the use of these exoskeletons are rarely mentioned.

Devices as prostheses are the most similar to upper limb exoskeletons. On these devices, adverse effects on health have been studied, particularly related to the biomechanics of the body and the distribution of loads during walking. It has been documented that while using a prosthesis can improve postural symmetry and facilitate certain functional movements, it can also induce compensations that compromise other musculoskeletal structures in the long term. The case of a child with absent radius syndrome who used a customized prosthesis, improving certain gait parameters, has been reported; however, the use of the prosthesis generated a significant overload in the left foot, increasing plantar pressure and contact time with the ground, which could lead to joint discomfort or overuse of the lower extremities over time [15]. In adults with unilateral amputations, it has been observed that the healthy arm tends to make wider movements to compensate for the loss of dynamic balance, which increases the risk of fatigue or overuse injuries [16]. Likewise, case studies such as the one presented by [17] highlight that the use of the prosthesis can alter trunk rotation and lateral flexion during walking, creating asymmetrical postural patterns. Without proper monitoring, these imbalances could affect stability and lead to musculoskeletal imbalances.

Figure 3 illustrates the contribution of the study and development of exoskeletons to implementing the Sustainable Development Goals (SDGs) proposed by the United Nations. Understanding the adverse effects on the musculoskeletal system of users provides criteria to improve design, performance, usability, and ergonomic conditions that drive better use of exoskeletons.

The development of affordable and low-cost exoskeletons contributes to the Sustainable Development Goals (SDGs) [18], especially in the areas of decent work and economic growth (SDG 8), industry, innovation, and infrastructure (SDG 9), and reduced inequalities (SDG 10). Making this technology more accessible promotes the inclusion of workers in sectors such as manufacturing, construction, and agriculture, reducing the risk of musculoskeletal injuries and improving working conditions, which favors a safer and more productive work environment. Designing affordable exoskeletons drives local innovation and collaboration across industries, fostering sustainable and resilient economies using more environmentally responsible materials and processes. These devices can benefit vulnerable communities, such as people with disabilities or low incomes, promoting equity and access to job opportunities and contributing to a more just and inclusive society. Exoskeletons are a key technology to assist people who are prone to severe injuries, strokes, or neurological diseases. These devices improve quality of life, health, and well-being (SDG 3), reduce physical limitations, and contribute to the health and safety of workers [19].

The design of exoskeletons contributes to the objective (SDG 9) since innovation contributes to industries’ infrastructure, allowing greater productivity and reducing energy consumption while providing ergonomic solutions to operators [20]. This effort also promotes research and collaboration between high- and low-income countries to design technologies tailored to local needs.

Using exoskeletons makes it possible to reduce inequalities in the height and strength of operators, promoting the reduction in inequalities (SDG 10). By focusing on accessibility and low-cost technologies, exoskeletons can be designed for resource-limited communities, reducing inequalities in using these technologies [21] and further promoting gender equality (SDG 5).

Table 1 summarizes the multiple contributions of upper-limb exoskeletons to the implementation of the SDGs proposed by the U.N.

This manuscript provides an overview of current knowledge on the adverse effects of exoskeleton use in work environments, identifying physiological, ergonomic, and technical risks. This study identifies and highlights muscle overload using lower-limb exoskeletons. However, it has limitations related to the methodological heterogeneity of the included studies, the predominance of simulated environments, the exclusion of non-indexed literature, and the lack of analysis of individual differences among users, which restricts the generalizability of its findings.

## 2. Methodology

The scoping review was carried out following the guidelines of the PRISMA^®^ methodology; more details of the review can be consulted in [35]. The search for information was conducted in scientific literature published in the last ten years. Journal and conference articles published in databases and repositories indexed in SCOPUS, Web of Science, ScienceDirect, Taylor & Francis, and PubMed have been considered.

The main question on which the research was based was, What are the effects on the musculoskeletal system due to the prolonged use of upper-limb exoskeletons? The questions used for the extraction of the information were RQ1. What effects have been evidenced due to the use of upper-limb exoskeletons? RQ2. What potential risks are associated with the prolonged use of upper-limb exoskeletons in the workplace? RQ3. How do upper-limb exoskeletons impact workers’ physical exertion and fatigue levels during repetitive tasks? RQ4. What ergonomic considerations need to be implemented in exoskeletons to assist workers in various industries? RQ5. What limitations exist in implementing solutions to reduce the negative effects of exoskeleton interaction and users? The review was carried out in three phases: elaborating the research questions, defining the scope, and planning an exhaustive search to collect all the relevant documents from which the information was extracted.

The exploratory review checklist proposed by PRISMA^®^ (Table A1 in Appendix A) was used, specifying the number of pages on which relevant information can be found. The questions described in Table 2 were applied to evaluate the quality of the selected scientific articles.

Figure 4 illustrates the process for selecting reference documentation related to the search keywords: “effects upper-limb exoskeleton workers”.

### 2.1. Inclusion Criteria

For this review, we exclusively selected peer-reviewed scientific articles and conference papers evaluating the use of upper-limb exoskeletons (ULEs) in occupational settings. Preference was given to those documents that analyze or address descriptively and explicitly the adverse or musculoskeletal effects that may arise from their prolonged or repetitive use. This includes alterations in muscle activation patterns, postural imbalances, muscle fatigue, and physical discomfort reported by users. Studies that presented empirical findings, either through biomechanical measurements, subjective assessments (e.g., user-reported pain or discomfort), or systematic reviews that addressed both the positive and negative effects of the use of upper-limb exoskeletons, were eligible. We included research applied in industrial contexts and controlled environments as long as they provided evidence on the possible adverse effects of these devices. The search strategy and the terms used are summarized in Table 3 to guarantee the transparency and reproducibility of the methodological process.

### 2.2. Exclusion Criteria

All those documents that addressed only the positive effects of the use of exoskeletons were excluded from this review, without considering or mentioning possible negative consequences, ergonomic limitations, or musculoskeletal impacts derived from their prolonged use, thus avoiding a biased view of the phenomenon and prioritizing studies with a more focused focus on adverse effects. Research focused exclusively on physical rehabilitation, motor recovery, or assisted therapy was discarded, as the main objective of this review is the use of upper-limb exoskeletons in occupational and industrial contexts. While these studies may offer valuable medical inputs, they do not align with the occupational risk assessment and musculoskeletal adverse effects approach in healthy workers.

Studies that focused on mathematical modeling of human movement or the development of computer simulations were excluded as they do not provide direct empirical evidence on the actual physiological effects of exoskeleton use. Similarly, research focused solely on methodologies for evaluation, technical validation, or prototyping of exoskeletons was discarded without analysis of their physical impact on human users.

## 3. Results

This section addresses aspects related to the effects associated with the interaction of exoskeletons and users in the work environment. Physical health risks, risks associated with prolonged use of exoskeletons, and ergonomic considerations that can improve the performance of exoskeleton use are addressed; finally, considerations and limitations in the implementation of solutions that minimize adverse effects and risks addressed are described.

### 3.1. Negative Effects Due to the Use of Upper-Limb Exoskeletons in the Work Environment

The use of upper-limb exoskeletons has shown significant benefits in reducing muscle load; however, adverse effects have also been evidenced that are important to consider for design, health, occupational safety, and ergonomics issues in developing these devices. This section describes the main negative effects evidenced in the scientific literature (Figure 5).

Several studies have documented the effects of the use of exoskeletons on the stability and balance of the user, particularly in dynamic or unstable contexts. This is because the structure of the device can change the body’s center of gravity, which increases the risk of tripping, especially in tasks that involve prolonged handling of loads [36]. In addition, in certain models exhibiting kinematic instability, limited benefits have been observed in reducing shoulder muscle activity, while a negative impact on the user’s overall balance has been reported [37].

Some research also points to an increase in muscle coactivation and an overload in other muscle groups [38]. Specifically, rigid designs can induce increased activation of antagonist muscles, such as the triceps brachii, which is activated to stabilize the shoulder joint [39]. Likewise, there has been an increase in extensor muscle activity, which overcompensates for antigravitational support during downward movements [40]. While load transfer to the lower extremities can relieve the shoulders, it also increases joint tension in the legs, which could lead to discomfort with prolonged use [41].

A decrease in the efficiency of dynamic movements has been observed when using upper-limb exoskeletons. In high-frequency repetitive activities, these devices can limit the natural range of motion and increase the user’s perceived exertion [40,42]. In addition, during evaluations carried out in real work environments, the benefits obtained in controlled environments—such as the 46% reduction in upper trapezius activity—are not always replicated, highlighting the need to improve the design for application in real contexts [43].

Other important aspects are related to ergonomics and customization of the device. Improper adjustments or incorrect dimensions can pressure sensitive areas, such as the wrists, leading to physical discomfort [44,45]. This discomfort, especially during prolonged use, has had a negative impact on device acceptance and user performance [44].

It has also been observed that positive laboratory results are not always maintained in real contexts, where effectiveness may depend on the user experience and specific operating conditions [45]. In tasks such as repetitive cargo handling in logistics, additional validations are required to understand its effectiveness in complex activities better [46].

In specific applications, such as cleaning surgical instruments, the use of the exoskeleton has contributed to reducing the load on muscles such as the anterior deltoid and latissimus dorsi; however, an increase in the activity of the erector spinal muscle has been reported, which can lead to back discomfort [44]. Similarly, in the textile sector, load redistribution from the shoulders to the lumbar area has been evidenced [47].

Passive devices, in particular, are not always adequately adapted to the diversity of tasks or anatomical differences between users. The rigidity of its structure can limit muscle flexibility and restrict mobility during complex tasks [48]. In addition, in activities with specific postures, such as repetitive arm movements, the level of support is not always optimally adjusted, which can lead to localized efforts in certain regions of the arm [49].

Perceptions of comfort and effort can vary widely among users due to individual differences and device configurations [43]. In some cases, users feel that the benefits gained do not outweigh the constraints of the design, which may affect their willingness to use it continuously [50].

Prolonged use without proper adjustments can lead to pressure build-up in areas such as the trunk and hips, underscoring the importance of ergonomic redesigns [7,51]. In industrial environments, certain passive models have not provided the necessary support to effectively reduce the effects of repetitive tasks over long working hours.

On the other hand, psychological risks associated with continuous use have also been identified, such as a 33% increase in cognitive load during material handling tasks [1]. The efficiency of the device can be limited if there is not adequate synchronization between the device and the user. In logistical activities, it has been observed that assistance applied directly to the wrist can generate additional stresses, hindering a natural interaction [46]. Likewise, if the assistance levels are not adjusted correctly, the user may be forced to exert additional effort, especially during downward arm movements [40]. Table 4 summarizes other less frequent negative effects reported in the scientific literature.

Empirical evidence has been obtained on the effects of passive exoskeletons on muscle activity during work tasks that require arm elevation [54]. The performance of the Skelex 360 exoskeleton in three industrial tasks (upper assembly, brick placement, and box movement) has been evaluated, finding significant reductions in the activation of shoulder flexor muscles such as the anterior and middle deltoids, especially in tasks above shoulder level, reaching reductions of up to 45.46% in the non-dominant anterior deltoid. Increases in extensor muscles such as the latissimus dorsi and posterior deltoids were reported in tasks performed below 90°, suggesting a redistribution of muscle effort according to posture and type of task.

The same exoskeleton was used in a controlled environment [4], and a dynamic painting simulation task with different arm elevation angles (60°, 90°, 120°, and 150°) was performed. A significant reduction in electromyographic (EMG) activity of the anterior and medial deltoid muscles was observed at all angles, with maximum reduction values of 38.2% and 38.7%, respectively. In this evaluation, no significant increases in the activation of other muscle groups were evidenced, suggesting a greater efficiency of the exoskeleton in dynamic tasks of continuous lifting.

The findings of the above-cited studies on the activation and inactivation of muscles associated with the use of upper-limb exoskeletons are illustrated in Figure 6.

### 3.2. Potential Risks Associated with Prolonged Use of Exoskeletons in the Workplace

Evaluations of upper-extremity exoskeletons have identified several risks associated with their use. These risks are mainly related to increased muscle activity, mobility restrictions, load transfer, improper posture, and an unbalanced distribution of the center of mass in the user-exoskeleton system.

One of the most relevant factors is increased muscle activity and overload, which pose considerable risks during exoskeleton use (Figure 6). These effects can lead to discomfort or long-term problems, especially when the muscle groups responsible for stabilizing the shoulder joint are overactivated [52]. This is most often observed during downward movements, where support overcompensation occurs [45], and also when the harness design does not favor adequate load distribution [46].

Increased muscle activity during work tasks, particularly when they involve high levels of strength and repetition, significantly increases the risk of developing musculoskeletal disorders (MSDs). Effects identified include inflammation in muscles, tendons, and bones; tissue breakdown; loss of bone quality; and joint damage. Additionally, neuromuscular alterations such as spinal cord sensitization, decreased grip strength, and the appearance of mechanical allodynia have been reported [55].

These effects are linked to cumulative fatigue and structural degradation, highlighting the importance of properly managing working conditions to mitigate these risks. Figure 7 illustrates the main dangers associated with increased muscle activity.

The mobility restrictions associated with using certain exoskeletons represent a relevant risk, as they limit the user’s natural range of motion due to the rigid design of some devices. This limitation compromises the worker’s comfort and can increase the perception of effort during specific tasks. In particular, shoulder mobility has been documented to be impaired in prolonged or dynamic activities, especially when the exoskeleton design does not fit the anatomical needs of the wearer [48,49]. This situation is exacerbated if the device does not offer adaptive support, which can significantly hinder work performance [24,36].

Another risk factor is the improper transfer of loads due to incorrect postures. This transfer type can stress other parts of the body, such as the lower extremities. Joint torques and leg discomfort have increased when the load is shifted to these areas [12]. In some cases, postures such as hyperextension of the knees have been linked to poor lumbar support design, which increases the risk of discomfort and injury [51].

Imbalance of the user-exoskeleton system can also compromise safety, especially in tasks that require movement. It has been observed that anteroposterior instability during load walks can increase the likelihood of tripping [20]. In addition, this imbalance can alter the displacement and natural speed of the body’s center of pressure, affecting the user’s overall balance [37].

The discomfort perceived by the user is another important risk factor. In the case of upper-extremity exoskeletons, localized discomfort has been identified in areas such as the wrist and shoulder [40]. Individual differences in device use and configuration also affect the system’s comfort and effectiveness due to variability in users’ anthropometric characteristics [47].

### 3.3. Impact of Using Exoskeletons During Repetitive Tasks

Using exoskeletons in repetitive and sustained tasks can generate adverse effects that must be carefully evaluated to optimize both their design and application. Among the impacts identified are inconsistencies in the reduction in muscle fatigue, variations in perceived physical exertion and productivity, effects on mobility, effectiveness according to the task, and user acceptance. These effects have been classified into physiological, psychological, and technological dimensions to provide a better understanding.

#### 3.3.1. Physiological Effects

Although upper-limb exoskeletons (ULEs) have demonstrated ergonomic benefits, several studies have also identified adverse physiological effects that should be considered in their implementation. First, while a significant decrease in muscle activity in muscles such as the deltoid has been reported during tasks above shoulder level [56,57], a compensatory increase in the activity of antagonist muscles, such as the triceps brachii and tibialis anterior, has been observed, which could lead to muscle imbalances in certain tasks [54]. The effectiveness of the support provided by ULEs varies considerably depending on the posture of the arm and the type of task; for example, in activities that do not require elevated movements, there may be an increase in extensor muscle activation [58].

From a cardiovascular perspective, some studies have shown an increase in cardiac cost and oxygen consumption during specific tasks, suggesting increased metabolic demand [56]; However, in repetitive tasks above shoulder level, reductions in cardiorespiratory response have also been documented [4], reflecting behavior highly dependent on the context of use.

In terms of postural effects, ULEs can induce postural tensions that alter movement patterns and generate discomfort [36]. Although they do not significantly affect gait parameters, some studies point to a decrease in anteroposterior stability, which could increase the risk of trips and falls. On the other hand, although many users report a reduction in the perception of physical exertion [7], discomfort related to improper adjustments, limitations in degrees of freedom, and usability issues can negatively affect the user experience and limit its overall effectiveness [59]. These findings highlight the need to improve exoskeletons’ ergonomic and functional design, considering greater customization and adaptation to specific tasks to mitigate their adverse effects [45].

#### 3.3.2. Psychological Effects

The use of upper-limb exoskeletons can generate adverse psychological effects that impact the acceptance and sustainability of the device in real contexts. One of the main challenges identified is cognitive-motor interference, as operating an exoskeleton requires additional cognitive resources that can affect the user’s overall performance and lead to mental fatigue [60].

Likewise, discomfort during use and usability issues have been reported to be significant barriers to widespread adoption, with women being particularly susceptible to these difficulties, underscoring the need for more inclusive approaches in design [61].

Another relevant factor is the relationship between body mass index (BMI) and habitual physical activity with the level of pain experienced when using these devices, which can lead to frustration, rejection, or decreased motivation to continue using them [62].

In addition, stability problems and associated postural tensions can increase the risk of falls, generating anxiety or fear of use, which has a negative impact on the user’s psychological well-being [36,58]. These aspects are compounded by potential gender differences in the perception of comfort and ease of use, reinforcing the need to consider individual variability in future design and implementation stages [60]. These findings highlight the importance of incorporating psychological assessments into exoskeleton validation, as well as developing more ergonomic and adaptable solutions that promote user confidence, comfort, and safety.

#### 3.3.3. Adverse Effects on Technology

Various studies have shown adverse effects from the technological point of view, which must be considered in their design and implementation. In terms of stability, an increased risk of tripping has been identified due to the alteration of anteroposterior stability, although the general gait parameters and the risk of slipping do not show significant changes [54]. In addition, using ULEs can modify the upper-limb kinematics and affect postural balance, increasing the risk of falls during manual handling tasks [63]. From a biomechanical point of view, although ULEs reduce muscle activity in muscles such as the deltoid, they can increase activation in other groups, such as the triceps brachii or tibialis anterior, which can lead to localized fatigue and physical discomfort. Likewise, cardiovascular demand has been reported during certain tasks, evidenced by increased heart rate and oxygen consumption [61]. In relation to user experience, although many report a lower perception of exertion [64], overall acceptance is affected by physical discomfort, unwanted muscle fatigue, and the need for task-specific adjustments [65]. Usability issues, such as difficulty in positioning the device, improper fit, or physical volume of the exoskeleton, also represent relevant obstacles to its adoption. In terms of specific applications, it has been observed that although ULEs are effective in tasks above shoulder level, their effectiveness decreases in other postures, where they can even induce greater activation of extensor muscles [66].

Among the main problems affecting technological development are alterations in joint mobility, postural imbalances, and increased load in regions such as the shoulders or spine, especially when the devices are not correctly adapted to the user. In addition, it has been observed that some models, being designed for specific tasks, can limit the ability to adapt to complex movements, affecting the worker’s efficiency and increasing the risk of compensatory injuries [50,67].

Despite the benefits in reducing muscle activity and preventing musculoskeletal disorders, there is a lack of long-term studies looking at the cumulative physiological effects of continued use. Perceptions of discomfort and stiffness during prolonged use have also been reported, which can negatively affect the acceptance of the device in the work environment. These findings underscore the importance of further developing exoskeletons with better ergonomics, adaptability, and validation in real-world working conditions [68].

Adverse events such as localized pain or skin lesions have been documented, underscoring the need for constant clinical monitoring and design improvements to ensure user safety. These findings show that the technological development of exoskeletons should focus on improving adaptability, dynamic ergonomics, and safe integration into various work tasks.

#### 3.3.4. Adverse Effects on Productivity and Task Completion

The diversity in exoskeleton designs entails variable effectiveness depending on the type of task and the environment in which they are used [1,48]. While significant reductions in muscle fatigue have been reported in controlled environments, increases in this same variable have been observed in real working conditions [43]. In repetitive over-the-shoulder tasks, decreased fatigue does not always translate into efficiency improvements in other muscle areas, which can limit the overall impact of the device [46].

The effects of perceived physical exertion and productivity can be contradictory. Some studies have shown a noticeable reduction in muscle activity in specific tasks [50]; however, the benefits related to the perception of effort and performance have been moderate [47]. In particular applications, such as high-frequency repetitive tasks, an increase in the time needed to complete activities has been evidenced, which counteracts the advantages in reducing effort [38]. In addition, in certain tasks that require controlled shoulder movements, a negative impact on the user’s overall balance has been reported [21].

In sectors such as logistics and transportation, exoskeletons do not always offer the support needed to handle varying loads [40]. The height and position required by certain tasks significantly influence the level of physical exertion and the device’s effectiveness, highlighting the importance of adjustable and customizable designs [42]. However, in activities such as lifting overhead loads, the devices have shown more consistent benefits, such as reduced fatigue and improved ergonomics [41]. On the other hand, the benefits have been less evident in dynamic or repetitive tasks with different configurations, which underlines the need for specialized designs depending on the application [7].

Comfort and adaptability to the device are influenced by the type of design and the degree of customization, factors that can increase the perceived effort and time needed to adapt to certain tasks [38]. Interaction between the user and the device in real-world environments is not always optimal, limiting the acceptance of the exoskeleton in the workplace [69]. In this sense, devices designed for specific tasks, such as logistics or cleaning surgical instruments, have shown better performance compared to generic designs, which tend to offer less consistent benefits [45].

The effects on productivity and efficiency in the work environment depend, to a large extent, on the characteristics of the task. This highlights the need to continue developing research that more robustly supports decision-making on adopting exoskeletons in organizations [48]. Multiple factors have also been identified that influence the acceptance and usability of the device, providing a useful basis for understanding the potential effects and risks of exoskeleton use on productivity and efficiency in the workplace [51].

### 3.4. Alternative Solutions to Reduce Adverse Effects on Exoskeletons

#### 3.4.1. Ergonomic Aspects

The ergonomic design of upper-extremity exoskeletons is a key factor in maximizing their effectiveness and reducing negative impacts on the wearer. Various studies have pointed out fundamental considerations for achieving a more efficient and adaptive design. These include customized anthropometric adjustments, load and force distribution mechanisms, improvements in freedom of movement and balance, customization of the torque level in actuators, and the integration of emerging technologies and specific solutions to improve usability in long shifts [49,52].

The possibility of making dynamic adjustments during use is a recommended strategy to increase the user’s flexibility and efficiency in varied work tasks [44]. It has been proposed that the incorporation of elements that optimize adaptation time, especially in repetitive activities, contributes to greater user acceptance of the device [38].

Regarding the distribution of physical efforts, it has been identified that an adequate load transfer from the back and shoulders to the hips can be achieved through appropriate design and specific components [41]. Likewise, developments have been proposed that reduce the impact on non-assisted areas, such as the lower extremities, to offer a more comfortable and safer user experience [40].

Another important aspect of the design is the minimization of restrictions on the user’s natural movement, allowing them to adopt postures and perform gestures necessary for the task [36] and maintain stable balance conditions. In this regard, the importance of developing exoskeletons that can be used in unstable environments, such as elevated platforms [45], has been highlighted. Likewise, it is suggested that devices should be versatile enough to adapt to different flexion and extension angles, thus optimizing their performance in various work scenarios.

Comfort during prolonged use is also a determining criterion for the acceptance and effectiveness of the exoskeleton. Therefore, improvements have been proposed in the device’s support systems, which do not compromise the user’s mobility [7,53]. Optimizing the exoskeleton’s adaptability to long working hours with various working postures is a priority to improve performance in repetitive tasks, as this significantly increases its overall effectiveness [1].

Task-focused design reduces risks and adverse effects. In sectors such as logistics and manufacturing, devices must maximize convenience and efficiency in order to improve productivity [47]. In addition, control strategies that adjust movement according to user intent are critical to providing positive ergonomic experiences [43,51].

Design proposals have been developed focused on reducing the time and effort required to place and adjust the device daily, which allows for improving operational efficiency and reducing resistance to use by workers [38]. Simplicity in adjustment and support mechanisms promotes smoother integration with existing workflows [66].

The integration of advanced technologies represents a promising avenue for boosting exoskeleton performance. Using sensors that monitor the user’s effort and automatically adjust the level of assistance significantly improves the interaction between the device and the operator [12]. In addition, adaptive systems enable efficient responses to task-specific demands in real-time, maximizing flexibility and efficiency in dynamic work environments [50].

#### 3.4.2. Use of Artificial Intelligence

Developments of upper-limb exoskeletons with artificial intelligence techniques have shown an improvement in the performance of exoskeletons by allowing them to adapt in a personalized way and in real-time to the needs of the user and the environment [70]. Using advanced algorithms, AI can analyze biomechanical data, predict movements, and adjust exoskeleton assistance to ensure smooth and efficient operation [71]. AI has been used to improve control through brain-computer interfaces, optimize human-machine interaction, and intelligently manage energy consumption to extend battery life [72]. In medical applications, the AI used in exoskeletons facilitates adaptive rehabilitation programs, while in work or military settings, it optimizes physical performance while reducing fatigue. Thanks to their ability to continuously learn, exoskeletons can evolve to offer an increasingly efficient, intuitive, and collaborative experience [73]. Figure 8 shows the most relevant areas in which AI contributes to the development of exoskeletons.

### 3.5. Limitations in Implementing Solutions to Reduce the Negative Effects of the Exoskeleton and User Interaction

The limitations in the interaction between the user and the exoskeleton and the efforts to mitigate its negative effects have been addressed from various perspectives. Among the most relevant aspects are familiarity with the device, anthropometric differences, lack of long-term studies, the gap between laboratory and field results, scalability restrictions, limitations in balance and movement, and the absence of validations in specific contexts.

One of the main challenges is related to the familiarization process and learning curve. It has been reported that the lack of time and resources dedicated to user adaptation significantly affects the device’s effectiveness [52]. This variability in the experience of use highlights the need for designs that allow adequate adaptation time and appropriate methodologies for practice in repetitive tasks [38].

Currently, many devices do not have the level of customization necessary to respond to the specific needs of users. For this reason, it is essential to consider anthropometric differences and design solutions that include people with previous injuries or with continuous support requirements. Limitations have been identified in certain designs that do not incorporate dynamic adjustments, which increases the load on other muscle groups and reduces their effectiveness [41].

The paucity of longitudinal studies and field evaluations has limited understanding of the long-term effects of exoskeleton use. Few studies analyze the sustainability of the use of these devices over time and their impact on occupational health [47,49]. In addition, the lack of robust empirical evidence in real-world environments underscores the need to validate its efficacy in specific industrial applications [43].

A discrepancy has been documented between the benefits obtained in laboratory environments and the results in real work scenarios. The positive effects observed under controlled conditions are not always replicated in contexts with high variability and dynamic demands [12,40]. Therefore, it is essential to carry out tests under real conditions that allow their performance to be evaluated more accurately.

Scalability continues to be a major constraint to adopting exoskeletons across industries. The lack of adaptive configurations has negatively impacted the effectiveness of the device in various tasks and environments [44]. The variability in effectiveness between different models reinforces the need for flexible designs that fit specific tasks [48].

Regarding mobility, restrictions related to kinematics and range of motion still persist. Current configurations do not always respect the user’s natural movement [21,39], and in many cases, fail to adequately balance the reduction in muscle activity with the minimization of antagonist muscle coactivation [52].

Training and coaching also present significant challenges. A lack of knowledge has been reported on properly interacting with the device and adjusting it to maximize benefits and reduce risks [7]. The absence of adequate training can lead to incorrect use, generating discomfort, inefficiency, or even long-term injuries [51].

The universal usability of exoskeletons is limited by factors such as anthropometric differences and the particular conditions of certain workers, including previous injuries or variability in assigned tasks [50]. Lack of adaptability in design may exclude users with disabilities or uncommon physical characteristics [42].

Economic and operational barriers need to be considered. Initial acquisition costs, required adaptations in the work environment, and necessary training may represent an obstacle to its implementation in some organizations [45]. Lack of compatibility with existing workflows or suboptimal designs can negate the expected benefits in terms of productivity [53].

Cultural and psychological factors can influence workers’ acceptance of the exoskeleton in the work environment. Some users may perceive the device as unnecessary or intrusive, especially if they do not clearly understand its benefits [27,30]. Resistance to change in traditional work settings can slow down the widespread adoption of these technologies, even when their efficacy is scientifically backed [67].

## 4. Discussion

While exoskeletons represent a promising solution for reducing physical load and preventing musculoskeletal injuries, the documented negative effects underscore key aspects that need to be addressed in future developments. These include reducing mobility restrictions, improving ergonomics, and encouraging device customization. To balance perceived benefits and potential risks, it is essential to adopt a holistic approach that considers both operating conditions and individual user needs, enabling effective implementation in diverse and demanding work environments.

The scientific literature has emphasized the importance of conducting longitudinal studies in real working conditions [47,67] to generate criteria that optimize the design of new exoskeletons and ensure that their benefits do not imply additional risks during the execution of tasks. In this context, it has been recommended to improve the dynamic performance of devices through more advanced ergonomic solutions and the incorporation of user feedback analysis, as well as the application of emerging technologies such as artificial intelligence to adapt support to specific demands [46,50].

The effectiveness of exoskeletons varies widely due to the diversity of existing designs, user experience, and the type of task performed [43]. This variability, coupled with the lack of safety regulations and standards, can increase the risks of fatigue and discomfort in the long-term [53]. Therefore, the evolution of these technologies must be accompanied by exhaustive evaluations of the biomechanical and physiological impacts in different work contexts. The development of more adjustable devices adapted to each work environment and their validation in real scenarios are essential to maximize benefits and minimize adverse effects.

The design and development of exoskeletons must be enriched through collaborative approaches that integrate researchers, designers, and end-users. This synergy will allow technologies to adapt more effectively to the real demands of the work environment. To overcome the current limitations in the interaction between the user and the device, it is necessary to comprehensively address the technical, human, and organizational aspects involved.

The use of upper-limb exoskeletons in the workplace generates significant social impacts, especially in terms of well-being and inclusion, by improving workers’ health and quality of life. In addition, they favor including people with physical limitations by allowing them to participate in previously inaccessible tasks, thus promoting more equitable and diverse work environments [74]. However, these benefits depend on adequate implementation, user acceptance, and organizational policies that guarantee their ethical use and adaptability to the real needs of the worker [58].

Manufacturers should focus on creating ergonomic, adjustable, and adaptive devices that are accessible to a wide variety of users. At the same time, implementing effective training programs and standardizing safety designs and measures are fundamental steps to boost the success of these technologies in the workplace, where they are set to become key tools of modern work.

The integration of artificial intelligence (AI) into upper-limb exoskeletons is a promising step forward in mitigating the adverse effects associated with their prolonged use in work or rehabilitation settings. Various AI-powered features, such as sensory integration, real-time control, and motion prediction, allow you to dynamically adapt the level of assistance, reduce muscle fatigue, and avoid improper postures [69]. Optimized by intelligent algorithms, human-exoskeleton collaboration improves movement synchronization and decreases involuntary resistance that can lead to discomfort or injury [71]. Likewise, the detection of faults and the optimization of energy consumption help to prevent mechanical failures that compromise the safety of the user [70]. These functionalities suggest that the design of AI-assisted exoskeletons improves operational efficiency and can also play a critical role in preventing adverse effects, thereby increasing their acceptance and applicability in the workplace and clinic. A relevant aspect to mitigate adverse effects is the development of methodologies, techniques, and evaluation protocols that allow the effectiveness and safety of exoskeletons in the workplace to be measured more accurately [68]. This will allow for more representative and applicable results, contributing to a safer and more efficient adoption of these technologies.

## 5. Conclusions

Using exoskeletons in repetitive tasks presents benefits and challenges highly dependent on the environment and nature of the activities. In controlled environments, exoskeletons have been shown to significantly reduce muscle fatigue in specific tasks, such as those involving over-the-shoulder movements. However, the results are inconsistent under real working conditions, as some designs increase fatigue due to inadequate load distribution or restrictions on the user’s natural mobility. In high-frequency tasks, the devices can increase the time needed to complete activities, counteracting the initial benefits of reduced physical effort by highlighting the need to develop exoskeletons adapted to specific tasks and with the ability to adjust to variations in the environment and work demands dynamically.

To maximize the effectiveness of exoskeletons and minimize their negative impacts, their ergonomic design must consider customization for each user. Dynamic adjustments during use can improve flexibility and efficiency in various tasks while ensuring comfort on long working days. A proper design should include mechanisms that efficiently distribute the load from areas such as the shoulders and back to the hips, reducing stress on unsupported muscle groups. Devices should allow for natural movements, necessary postures, and tasks in unstable environments, improving balance and preventing harmful muscle coactivations. Integrating advanced technologies, such as sensors to monitor effort and adaptive systems to adjust the level of support in real-time, can optimize user-device interaction, improving both usability and acceptance of the exoskeleton.

Although exoskeletons show promising results in controlled environments, the lack of longitudinal research and evaluations in real work scenarios limits their effective implementation. Recent studies indicate that the benefits observed in laboratories, such as reduced fatigue and improved productivity, are not always replicated in real conditions due to factors such as high variability in tasks, dynamic demands, and anthropometric differences of users. This underscores the need for extensive field testing to assess the sustainability of device use, their long-term impact on occupational health, and their integration into existing workflows. In addition, developing design and safety standards and effective training programs can improve the acceptance and proper use of exoskeletons in various industries. By addressing these challenges, maximizing the perceived benefits and minimizing the risks associated with these technologies will be possible.

## Figures and Tables

**Figure 1 biomimetics-10-00340-f001:**
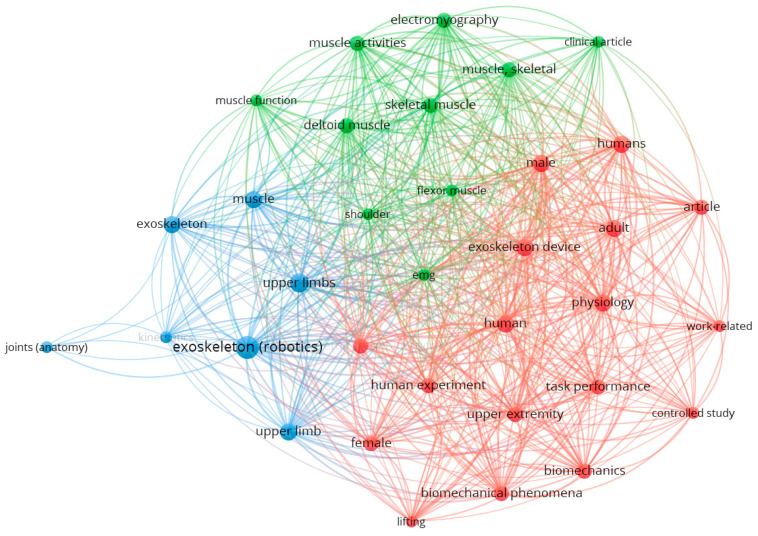
Bibliometric graph of articles related to the SCOPUS database that address the negative effects on the musculoskeletal system due to the interaction of using upper-limb exoskeletons. The graph was obtained with VOSviewer^®^ version 1.6.20.

**Figure 2 biomimetics-10-00340-f002:**
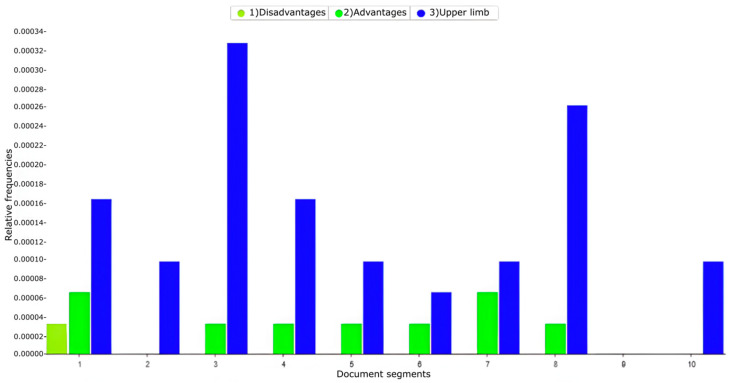
Relative frequency of studies addressing the advantages and disadvantages of using upper limb exoskeletons for industrial work.

**Figure 3 biomimetics-10-00340-f003:**
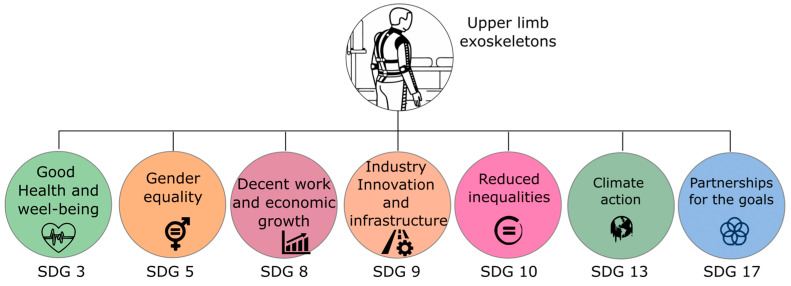
Contributions of the development of exoskeletons to the Sustainable Development Goals proposed by the UN.

**Figure 4 biomimetics-10-00340-f004:**
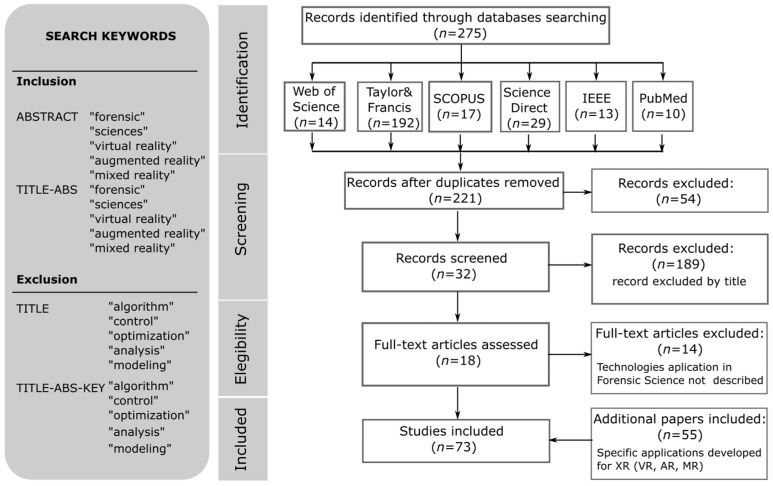
Workflow according to the guidelines of the PRISMA^®^ methodology.

**Figure 5 biomimetics-10-00340-f005:**
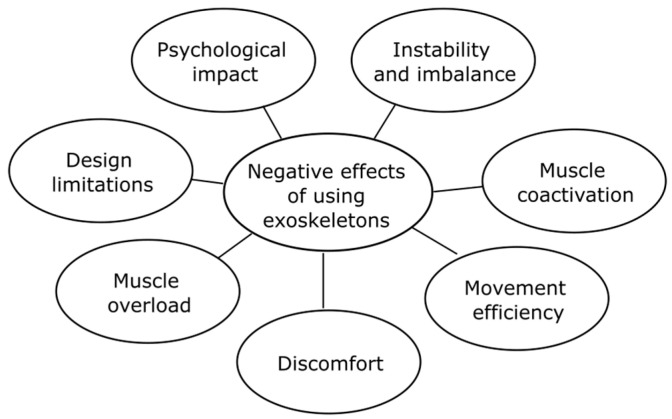
Negative effects of using exoskeletons in the workplace.

**Figure 6 biomimetics-10-00340-f006:**
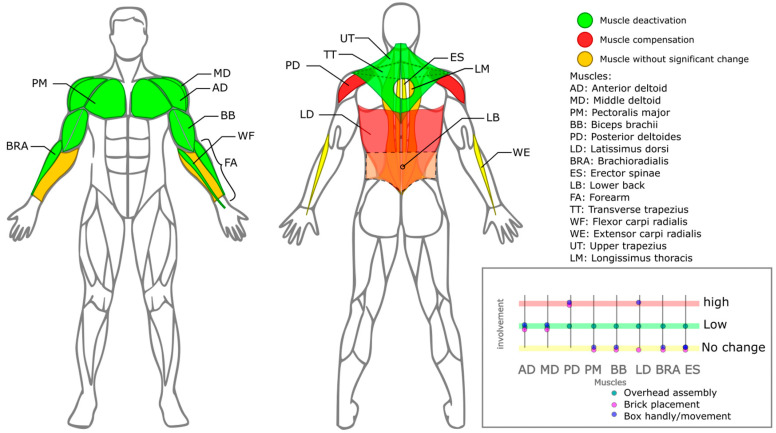
Compensation and deactivation of muscles during the use of upper-limb exoskeletons.

**Figure 7 biomimetics-10-00340-f007:**
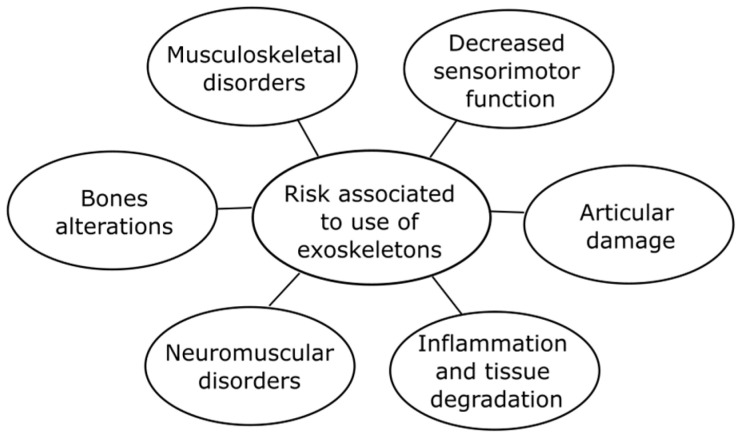
Risks associated with increased muscle activity due to prolonged use of exoskeletons.

**Figure 8 biomimetics-10-00340-f008:**
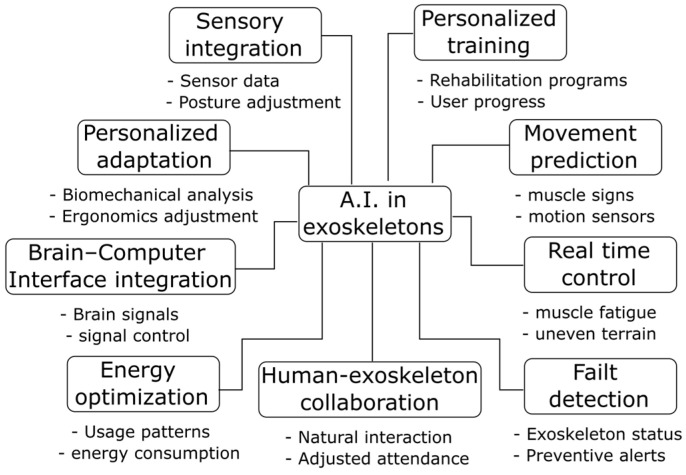
Artificial intelligence contributes to improving exoskeletons’ performance in the work environment.

**Table 1 biomimetics-10-00340-t001:** Contributions of upper-limb exoskeletons to the execution of the Sustainable Development Goals (SDGs).

SDG	Contribution/Issue	References
SDG 3: Health and Well-being	Rehabilitation technologies and assistive devices enhance quality of life for individuals with motor impairments.	[22,23]
SDG 3: Health and Well-being	Mitigates occupational musculoskeletal strain in rehabilitation therapists during patient sessions.	[24]
SDG 5: Gender equality	Promotes inclusion, accessibility, empowerment and equity in rehabilitation technologies.	[25]
SDG 8: Decent Work and Economic Growth	Supports industrial workers by reducing musculoskeletal risks and improving productivity.	[26,27,28]
SDG 8: Decent Work and Economic Growth	Enhances ergonomic safety and occupational health outcomes by reducing work-related injuries and improving job satisfaction.	[26,29,30]
SDG 9: Industry, Innovation and Infrastructure	Promotes technological innovation in advanced materials, intelligent control systems, and optimized human-machine interfaces.	[22,31]
SDG 9: Industry, Innovation and Infrastructure	Supports smart manufacturing and Industry 4.0 integration.	[30,32]
SDG 10: Reduced Inequalities	Enhances accessibility for persons with disabilities, advancing equality.	[33,34]
SDG 11: Sustainable Cities and Communities	Enhances mobility and independence, fostering inclusive communities	[23]

**Table 2 biomimetics-10-00340-t002:** Quality Assessment Questions for paper quality.

Quality Assessment Questions Answer	Answer
Does the document describe the effects associated with the prolonged use of upper-limb exoskeletons in work activities?	(+1) Yes/(+0) No
Does the paper describe the health risks associated with the prolonged use of upper-limb exoskeletons in the work environment?	(+1) Yes/(+0) No
Are ergonomic recommendations addressed or given to reduce workers’ health risks due to the use of exoskeletons?	(+1) Yes/(+0) No
Is the journal or conference in which the article was published indexed in the JCR?	(+1) if it is ranked Q1, (+0.75) if it is ranked Q2, (+0.50) if it is ranked Q3, (+0.25) if it is ranked Q4, (+0.0) if it is not ranked

**Table 3 biomimetics-10-00340-t003:** Strings are used by search for each database.

Database	String Search	Studies Number
Web of Science	effects upper limb exoskeleton workers (Topic)	35
Taylor & Francis	Abstract: effects upper limb exoskeleton workers	7
Science Direct	Title, abstract, keywords: effects upper limb exoskeleton workers	6
Scopus	TITLE-ABS-KEY (effects AND upper AND limb AND exoskeleton AND workers)	31
PubMed	search: effects upper limb exoskeleton workers	37
	Total number of studies	116

**Table 4 biomimetics-10-00340-t004:** Negative effects due to the use of exoskeletons on workers.

Negative Effect	Description	Ref
Discomfort and pain	Due to muscle weakness, inadequate redistribution of loads, restriction of movements, increase in body temperature, and fatigue due to lack of adaptation.	[52]
Increased cognitive load	It occurs because the worker must adapt to new movement patterns, divide their attention between the task and the device, and manage possible discomforts. This increases mental effort and fatigue, affecting efficiency and safety at work.	[36]
Task-specific constraints	Due to restrictions in movement, agility is reduced, making tasks that require flexibility or precision difficult, generating muscle fatigue in unassisted areas, and increasing discomfort in prolonged postures.	[42,53]
Negative perception and usability	It arises from discomfort, movement restrictions, and fatigue, which makes it difficult to use for a long time and can generate rejection by the worker.	[12]
Possibility of increased fatigue	It is due to overload in unassisted muscles, restriction of natural movements, increased cognitive effort, and possible discomfort in adjustment. The weight of the exoskeleton and the need to adapt to its functioning can lead to physical and mental exhaustion.	[42,52]
Ergonomic and design challenges	Lack of standardized methodologies for ergonomic analysis and optimization.	[7,48]

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
