# Peer review of "Adverse Effects Due to the Use of Upper Limbs Exoskeletons in the Work Environment: A Scoping Review"

_biomimetics, 2025, doi:10.3390/biomimetics10050340_

Round 1
Reviewer 1 Report
Comments and Suggestions for Authors
This paper reviewed the scientific literature on the negative effects that have been evidenced due to the use of upper limb exoskeletons. To further improve the quality of the paper, the following suggestions are put forward:
- This paper cited 46 literatures, with a total of 13 pages of text content, which is far from enough for a review paper. This hardly convinces the reader that the author has a thorough understanding of developments in the field.
- The abstracts of this paper only show that the authors collected relevant literature and analyzed it using PRISMA® method. This does not clearly state the main work content, conclusion and contribution of this paper. The last sentence of the abstract is the conclusion of the research paper, not the review paper.
- The logic of the introduction of this paper is not clear. It does not explain the research significance of this paper, the main contribution, and the framework of the paper. In the introduction, the sentence "This scoping review seeks to community settings" summarizes the work to be carried out in the paper, but obviously does not match the content that the paper mainly studies the negative effects of the upper limb exoskeleton.
- There are no physical or scene pictures related to the upper limb exoskeleton and the negative effects of the upper limb exoskeleton.
- There are many formatting problems in the paper.
Author Response
REVIEWER 1
We sincerely thank the reviewer for their time and valuable comments, which have significantly contributed to improving the quality of our manuscript.
Comment 1
This paper cited 46 literatures, with a total of 13 pages of text content, which is far from enough for a review paper. This hardly convinces the reader that the author has a thorough understanding of developments in the field.
Response 1
The developmental aspects have been expanded in greater detail in terms of the description of the muscles that are deactivated and compensated when upper limb exoskeletons are used. The discussion has been enriched by addressing the contributions of AI in better functioning and as a tool to reduce adverse effects on exoskeletons. The development and content of the article have been enriched and unmentioned with the contribution and comments of all the reviewers.
Comment 2
The abstracts of this paper only show that the authors collected relevant literature and analyzed it using PRISMA® method. This does not clearly state the main work content, conclusion and contribution of this paper. The last sentence of the abstract is the conclusion of the research paper, not the review paper.
Response 2
The abstract has been written better, highlighting the main content, the main conclusion around the findings of the search carried out, and the contribution of this manuscript.
Comment 3
The logic of the introduction of this paper is not clear. It does not explain the research significance of this paper, the main contribution, and the framework of the paper. In the introduction, the sentence "This scoping review seeks to community settings" summarizes the work to be carried out in the paper, but obviously does not match the content that the paper mainly studies the negative effects of the upper limb exoskeleton.
Response 3
The introduction has been organized and written better. The main problem around the lack of systematized studies in this field has been highlighted, and the contribution of this manuscript has been highlighted better. Texts that confused the reading have been removed. The framework and limitations of this study were included.
Comment 4
There are no physical or scene pictures related to the upper limb exoskeleton and the negative effects of the upper limb exoskeleton.
Response 4
The Figure 5 has been included to identify the deactivation and compensation of the muscles during the use of upper limb exoskeletons; it includes a comparison when the user performs three different tasks on the head. The description has been placed on lines 323-340.
Comment 5
There are many formatting problems in the paper.
Response 5
Fixed issues around formatting and using the template provided by the journal. References, fonts, and spacing of the entire document have been corrected.
Reviewer 2 Report
Comments and Suggestions for Authors
Through a systematic literature review, this article examines the negative impacts of upper limb exoskeletons on users in the workplace, particularly concerning the musculoskeletal system. The article is well-structured, employs a sound research methodology, presents detailed content, and possesses significant academic value. However, certain sections require further optimization, particularly regarding the depth of the literature review, the interpretation of the results, and the logical flow of the discussion section. Overall, the article is acceptable pending major revisions.
The current results section (Section 3) includes some discussion, such as an explanation of negative effects. It is recommended to separate the results from the discussion to enhance the organization of the article.
The discussion section could benefit from greater depth, particularly in the interpretation of the findings and the suggestions for future research directions. For instance, the technical challenges associated with exoskeleton design—such as material selection and power system optimization—should be examined more thoroughly. Additionally, the potential for enhancing the adaptability of exoskeletons through artificial intelligence techniques warrants further exploration.
The article outlines several negative effects, such as muscle fatigue and cognitive load; however, it lacks a systematic categorization of these effects. It is recommended to classify them according to physiological, psychological, and technological dimensions to enhance the reader's understanding. While the article acknowledges the absence of long-term studies, it does not explore the underlying causes of this issue or its implications for the findings. It is advisable to include pertinent content in the discussion section.
The Sustainable Development Goals (SDGs) section of Figure 2 could be further optimized. For instance, incorporating more specific examples or data would effectively illustrate how exoskeleton technology directly contributes to the achievement of the SDGs.
The literature cited in the article primarily focuses on the period before 2020. It is recommended to include the latest research from recent years, particularly regarding the application of artificial intelligence in exoskeletons. Additionally, the references are not uniformly formatted, and the reference numbers do not correspond to the citations in the text. It is advisable to standardize the format in accordance with the journal's requirements.
Comments on the Quality of English LanguageThe English could be improved to more clearly express the research.
Author Response
REVIEWER 2
We sincerely thank the reviewer for their time and valuable comments, which have significantly contributed to improving the quality of our manuscript.
Comment 1
Through a systematic literature review, this article examines the negative impacts of upper limb exoskeletons on users in the workplace, particularly concerning the musculoskeletal system. The article is well-structured, employs a sound research methodology, presents detailed content, and possesses significant academic value. However, certain sections require further optimization, particularly regarding the depth of the literature review, the interpretation of the results, and the logical flow of the discussion section. Overall, the article is acceptable pending major revisions.
Response 1
The results section has been broadened and deepened by including studies that highlight muscle deactivation and compensation using upper limb exoskeleton and in performing common tasks such as load handling and overhead assembly. Figure 5 illustrating these muscle areas has been included followed by their description. Additionally, the discussion section has been improved to provide a more fluid reading; the contributions of AI to reduce the adverse effects of the use of these devices have been discussed.
Comment 2
The current results section (Section 3) includes some discussion, such as an explanation of negative effects. It is recommended to separate the results from the discussion to enhance the organization of the article.
Response 2
Document writing has been improved by omitting aspects that can be placed in the discussion.
Comment 3
The discussion section could benefit from greater depth, particularly in the interpretation of the findings and the suggestions for future research directions. For instance, the technical challenges associated with exoskeleton design—such as material selection and power system optimization—should be examined more thoroughly. Additionally, the potential for enhancing the adaptability of exoskeletons through artificial intelligence techniques warrants further exploration.
Response 3
The wording of the discussion has been improved to give it greater depth, future contributions to mitigate the adverse effects of the use of these exoskeletons are discussed. The contribution of AI in improving this aspect of exoskeletons is addressed.
Comment 4
The article outlines several negative effects, such as muscle fatigue and cognitive load; however, it lacks a systematic categorization of these effects. It is recommended to classify them according to physiological, psychological, and technological dimensions to enhance the reader's understanding. While the article acknowledges the absence of long-term studies, it does not explore the underlying causes of this issue or its implications for the findings. It is advisable to include pertinent content in the discussion section.
Response 4
The physiological, psychological and technological effects have been included in point 3.3 on Impacts of exoskeleton use during repetitive tasks and relevant references to the above topics have been included. The introduction addresses the problem of the lack of long-term studies on these adverse effects.
Comment 5
The Sustainable Development Goals (SDGs) section of Figure 2 could be further optimized. For instance, incorporating more specific examples or data would effectively illustrate how exoskeleton technology directly contributes to the achievement of the SDGs.
Response 5
Despite not obtaining precise statistical values that justify each contribution to the SDGs, this aspect has been enriched with the inclusion of a table highlighting the specific contributions to the aforementioned SDGs.
Comment 6
The literature cited in the article primarily focuses on the period before 2020. It is recommended to include the latest research from recent years, particularly regarding the application of artificial intelligence in exoskeletons. Additionally, the references are not uniformly formatted, and the reference numbers do not correspond to the citations in the text. It is advisable to standardize the format in accordance with the journal's requirements.
Response 6
Relevant bibliography from the last four years on the content addressed has been included in the results section, above all in the results section.
Reviewer 3 Report
Comments and Suggestions for Authors
The study has a solid foundation and makes an important contribution to developing knowledge regarding using exoskeletons. Certain extensions, particularly in quantitative data, field research, and the analysis of psychological aspects of use, could enhance its applicability and impact.
Main remarks:
- The article covers a wide range of topics related to exoskeletons, from their advantages, such as reducing muscle fatigue, to their limitations and negative effects. The article's subject matter is relevant and highly significant in the context of industrial development and improving working conditions.
- The article addresses an important topic but exhibits clear empirical and quantitative research gaps. An excessive focus on literature analysis limits the practical applicability of the findings. Furthermore, it would be valuable to deepen the reflection on the impact of exoskeletons within diverse social and psychological contexts.
- Despite mentioning the phases of the research process and the use of PRISMA®, the paper lacks a more detailed description of the data analysis, which hinders the evaluation of the accuracy of its conclusions. As a result, the findings are more narrative than statistical, reducing their credibility. The absence of tables with precise results and statistical analyses introduces a risk of overgeneralization.
- To increase the scientific value of the work, the analysis of the literature should also include studies on other movement-assisting devices, such as prostheses, and their impact on health aspects, including body posture or musculoskeletal system loads. The works addressing this topic are: [1] Yancosek, K. E., Schnall, B. L., & Baum, B. S. (2008). Impact of upper-limb prosthesis on gait: A case study. Journal of Prosthetics and Orthotics, 20(4). https://doi.org/10.1097/JPO.0b013e31818adb29, [2] Armstrong, K., Brinkmann, J. T., Stine, R., Gard, S. A., & Major, M. J. (2021). Do Upper-Limb Loss and Prosthesis Use Affect Lower-Limb Gait Dynamics? Journal of Prosthetics and Orthotics, 33(4). https://doi.org/10.1097/JPO.0000000000000333, [3] Glowinski, S.; Pecolt, S.; BÅ‚ażejewski, A.; Maciejewski, I.; Królikowski, T. Gait Analysis with an Upper Limb Prosthesis in a Child with Thrombocytopenia–Absent Radius Syndrome. J. Clin. Med. 2025, 14, 2245. https://doi.org/10.3390/jcm14072245. Including studies on prostheses could broaden the perspective of analysis, allowing for a comparison of the impact of different movement-assisting technologies on the functioning of users in everyday life and the work environment.
- Phrases such as "repetitive tasks" and "high-frequency tasks" are used interchangeably, which might cause slight confusion for the reader. It is advisable to standardize the terminology throughout the text.
- Some sentences are overly long and complex, which may affect the readability and clarity of the text.
- The text occasionally repeats ideas or concepts, such as the need for adaptation and ergonomics, without adding new information. Consolidating these points would improve conciseness.
Author Response
REVIEWER 3
We sincerely thank the reviewer for their time and valuable comments, which have significantly contributed to improving the quality of our manuscript.
Comment 1
The article covers a wide range of topics related to exoskeletons, from their advantages, such as reducing muscle fatigue, to their limitations and negative effects. The article's subject matter is relevant and highly significant in the context of industrial development and improving working conditions.
Response 1
It has been incorporated in sections 3.1 to Figure 5, which illustrates the deactivation and compensation of muscles due to the use of exoskeletons for overhead tasks. The description in Figure 5 addresses values of these effects in some muscles. The discussion addresses how the study of adverse effects can improve exoskeletons and contribute to productive aspects of the industrial environment. In addition, the impact of exoskeletons on the social context has been included in the discussion.
Comment 2
The article addresses an important topic but exhibits clear empirical and quantitative research gaps. An excessive focus on literature analysis limits the practical applicability of the findings. Furthermore, it would be valuable to deepen the reflection on the impact of exoskeletons within diverse social and psychological contexts.
Response 2
The document has been enriched with studies that address the deactivation and compensation of muscles due to the use of upper limbs exoskeletons and the impacts in the physiological, psychological and technological contexts. The discussion has been enriched by addressing the impact on the social aspect.
Comment 3
Despite mentioning the phases of the research process and the use of PRISMA®, the paper lacks a more detailed description of the data analysis, which hinders the evaluation of the accuracy of its conclusions. As a result, the findings are more narrative than statistical, reducing their credibility. The absence of tables with precise results and statistical analyses introduces a risk of overgeneralization
Response 3
Some experimental values from some studies have been included in the description in Figure 5. Baseline studies have not conducted long-term studies and evaluations on the impacts and adverse effects of using upper limb exoskeletons. To strengthen the methodology, the inclusion and exclusion criteria have been written in more detail, in addition to this, it is mentioned that citation number 35 has the details on the extraction of information carried out in a dataset published in Mendeleydata.
Comment 4
To increase the scientific value of the work, the analysis of the literature should also include studies on other movement-assisting devices, such as prostheses, and their impact on health aspects, including body posture or musculoskeletal system loads. The works addressing this topic are: [1] Yancosek, K. E., Schnall, B. L., & Baum, B. S. (2008). Impact of upper-limb prosthesis on gait: A case study. Journal of Prosthetics and Orthotics, 20(4). https://doi.org/10.1097/JPO.0b013e31818adb29, [2] Armstrong, K., Brinkmann, J. T., Stine, R., Gard, S. A., & Major, M. J. (2021). Do Upper-Limb Loss and Prosthesis Use Affect Lower-Limb Gait Dynamics? Journal of Prosthetics and Orthotics, 33(4). https://doi.org/10.1097/JPO.0000000000000333, [3] Glowinski, S.; Pecolt, S.; BÅ‚ażejewski, A.; Maciejewski, I.; Królikowski, T. Gait Analysis with an Upper Limb Prosthesis in a Child with Thrombocytopenia–Absent Radius Syndrome. J. Clin. Med. 2025, 14, 2245. https://doi.org/10.3390/jcm14072245. Including studies on prostheses could broaden the perspective of analysis, allowing for a comparison of the impact of different movement-assisting technologies on the functioning of users in everyday life and the work environment.
Response 4
The suggested studies have been increased in the introduction section, and prostheses have been addressed as devices that have been studied on their adverse effects, this to highlight the importance of studying adverse effects on exoskeletons.
Comments 5
Phrases such as "repetitive tasks" and "high-frequency tasks" are used interchangeably, which might cause slight confusion for the reader. It is advisable to standardize the terminology throughout the text.
Some sentences are overly long and complex, which may affect the readability and clarity of the text.
The text occasionally repeats ideas or concepts, such as the need for adaptation and ergonomics, without adding new information. Consolidating these points would improve conciseness.
Response 5
The clarity of the expressions and some sentences that could confuse the reader have been improved. The use of words that are not precise has been verified to improve the comprehension and clarity of the text. Repeated ideas have been removed as discussed.
Round 2
Reviewer 1 Report
Comments and Suggestions for Authors
After reading the revised paper, it is found that the author has made a lot of modifications to the previous review comments, and the format is more accurate and the logic is more rigorous. However, there are still places for further optimization to better meet the publication standards of this journal:
- The paper still has the drawback of insufficient references. The selection range of its analysis databases is limited (such as not including engineering databases like IEEE Xplore), which may lead to the results being biased towards the medical/biological perspective and lacking the analysis of technical design flaws. It is suggested that the author further expand the scope of literature retrieval and can refer to some industry reports on exoskeletons to reflect the deficiencies and challenges in practical applications.
- The visualization and data presentation of the current research status of the thesis are insufficient. The research status of the paper is mainly presented in the form of tables. Review papers should visually summarize the research status and key findings in the form of a large number of figures.
Author Response
REVIEWER 1
We sincerely thank the reviewer for their time and valuable comments, which have significantly contributed to improving the quality of our manuscript.
Comment 1
The paper still has the drawback of insufficient references. The selection range of its analysis databases is limited (such as not including engineering databases like IEEE Xplore), which may lead to the results being biased towards the medical/biological perspective and lacking the analysis of technical design flaws. It is suggested that the author further expand the scope of literature retrieval and can refer to some industry reports on exoskeletons to reflect the deficiencies and challenges in practical applications.
Response 1
The search has been carried out in the IEEE Xplore database, and some studies on adverse effects that have already been considered in the previous references have been identified, however, two studies have been added that provide details on adverse effects that hinder the implementation and development of design aspects for the development of these technologies.
Comment 2
The visualization and data presentation of the current research status of the thesis are insufficient. The research status of the paper is mainly presented in the form of tables. Review papers should visually summarize the research status and key findings in the form of a large number of figures.
Response 2
Figure 2 has been included, which highlights studies on exoskeletons of upper limbs obtained with the text string: ("upper limb" OR "arm" OR "forearm" OR "hand") AND ("exoskeleton" OR "wearable" OR "robotic" OR "assistive") AND ("industrial" OR "workplace" OR "manufacturing" OR "fac-tory") AND ("workers" OR " employees" OR "operators" OR "laborers") AND ("ergonomics" OR "safety" OR "performance" OR "fatigue") AND ("rehabilitation" OR "support" OR "augmentation" OR "augmentation" OR "enhancement"). The articles analyzed in Figure 2 correspond to the SCOPUS database. The figure highlights the relative frequency of terms about advantages and disadvantages of using exoskeletons since this aspect includes adverse, counterproductive, and negative effects. It is noted that the studies lack focus on the latter aspects, suggesting that this study provides a real knowledge gap.

Reviewer 3 Report
Comments and Suggestions for Authors
Thank you for considering my comments. I am glad that my suggestions were taken into account.
Author Response
REVIEWER 3
We sincerely thank the reviewer for their time and valuable comments, which have significantly contributed to improving the quality of our manuscript.
Comment 1
Thank you for considering my comments. I am glad that my suggestions were taken into account.
Response 1
We appreciate the time in reviewing our work and your comments that have allowed us to improve our manuscript.
